# Vaginal Nanoformulations for the Management of Preterm Birth

**DOI:** 10.3390/pharmaceutics14102019

**Published:** 2022-09-23

**Authors:** Asad Mir, Richa V. Vartak, Ketan Patel, Steven M. Yellon, Sandra E. Reznik

**Affiliations:** 1Department of Pharmaceutical Sciences, St. John’s University, Queens, NY 11439, USA; 2Department of Basic Sciences, Physiology Division, Loma Linda School of Medicine, Loma Linda, CA 92350, USA; 3Departments of Basic Sciences and Obstetrics and Gynecology, Longo Center for Perinatal Biology, Loma Linda School of Medicine, Loma Linda, CA 92350, USA; 4Department of Pathology, Albert Einstein College of Medicine, Bronx, NY 10461, USA; 5Department of Obstetrics and Gynecology and Women’s Health, Albert Einstein College of Medicine, Bronx, NY 10461, USA

**Keywords:** vaginal administration, preterm birth, progesterone, cervical ripening, uterine first-pass effect, vaginal nanoformulation

## Abstract

Preterm birth (PTB) is a leading cause of infant morbidity and mortality in the world. In 2020, 1 in 10 infants were born prematurely in the United States. The World Health Organization estimates that a total of 15 million infants are born prematurely every year. Current therapeutic interventions for PTB have had limited replicable success. Recent advancements in the field of nanomedicine have made it possible to utilize the vaginal administration route to effectively and locally deliver drugs to the female reproductive tract. Additionally, studies using murine models have provided important insights about the cervix as a gatekeeper for pregnancy and parturition. With these recent developments, the field of reproductive biology is on the cusp of a paradigm shift in the context of treating PTB. The present review focuses on the complexities associated with treating the condition and novel therapeutics that have produced promising results in preclinical studies.

## 1. Introduction

With about 15 million babies born preterm each year, preterm birth (PTB), defined as birth before 37 completed weeks of gestation, is the leading cause of neonatal morbidity and mortality worldwide [1,2,3]. According to the World Health Organization, the annual rate of PTB has not improved over the last several decades and exceeds 10% of all pregnancies in most countries [4]. Interestingly, the United States (US) has the highest rate of PTB among industrialized nations [2]. Women in the US have high rates of obesity and cardiovascular disease and are subjected to the social stressors of a diverse population, such as racism and income disparity. The high rate of PTB in the US may reflect a complex interaction of these socioeconomic and co-morbidity factors [5].

Children born prematurely have an increased risk of developing neurological, respiratory, and metabolic abnormalities, and effects on cognitive function are often lifelong [6]. The costs associated with PTB, both acute and lifelong, have broad and sustained effects on families and are enormously expensive for society. In the developed world, advances in neonatology have outpaced advances in the PTB field, so that more and more infants at the cusp of viability survive their neonatal intensive care unit stay, only to face a constellation of lifelong medical challenges. Unfortunately, to date, the development of safe and effective drug therapy to prevent PTB has been impeded by the seemingly unsurmountable obstacles of (1) concern for teratogenic effects on the developing fetus, (2) concern for toxic effects on the mother, and (3) lack of convincing evidence of improved neonatal outcomes. In this article, we present a systematic review, based on our survey of peer-reviewed literature from 1979 to the present, focused on vaginal nanoformulations as a new approach to overcome these challenges.

## 2. Impediments to Treat PTB

Compared to other clinical problems, PTB has not fully benefitted from advances in technology and practice in modern medicine over the last century. Numerous complex etiologies may contribute to the syndrome [7,8,9,10]. Assessing risk factors, demographics of the patient, biomarkers, and cervical length aids in the screening and management of preterm labor [11,12,13], but the challenge remains to determine accurately if or when preterm labor will occur. Bibliometric analyses show an exponential world-wide increase in research in this field [14]. Until recently, in some clinical settings, cervical cerclage was the main treatment option for the diagnosis of incompetent or short cervix, but the effectiveness of this approach is unclear [13,15]. Currently, there are two recommended treatment options for the management of short cervix (<25 mm) occurring between 16 and 24 weeks’ gestation in the US: cerclage and vaginal progesterone. 

### 2.1. Progesterone Reduces Risks of Preterm Labor and Birth

The role of progesterone in maintaining pregnancy cannot be overstated. Although many hormones are important regulators of physiological processes during pregnancy, only progesterone can sustain reproductive structures that support fetal development until term in mammals [16]. Questions about the positive effects that progesterone has in the prevention of PTB may not extend to the variety of risk phenotypes for spontaneous premature conclusion of pregnancy [17]. Among many physiological roles, one of the major mechanisms by which progesterone maintains pregnancy is to inhibit contractions of the myometrium [18,19]. This effect is accomplished by changing the resting potential of smooth muscle cells via suppressing the calcium–calmodulin–myosin light chain kinase system [20,21]. In pregnant rats, progesterone blocks inflammatory changes in cervix structure and collagenolysis before PTB [16]. In both rats and mice, specific progesterone receptor agonists or antagonists regulate cervix remodeling associated with prevention or promotion of PTB, respectively [22,23].

At term, cross-species comparisons indicate that progesterone in circulation remains high in pregnant rodents prior to labor, when the myometrium transitions to a contractile phenotype a day or so before birth [24,25]. Thus, in these species during the period of cervix ripening 3–5 days before birth, systemic progesterone is at or near peak concentrations. By comparison in human pregnancy, progesterone concentrations remain elevated and continue to increase throughout pregnancy until birth. These findings support the hypothesis that pregnant rodents and women undergo a functional withdrawal with loss of response to progesterone receptor-mediated effects for cervix ripening and for labor in women [7,26]. How in vitro findings about the role for progesterone receptor (PR) subtypes as pro- or anti-inflammatory mediators or the ratio of PR-A and PR-B isoforms [20,27,28] translate to the parturition process requires further study. Indeed, mice lacking the PR-B subtype do not have impaired pregnancy, cervix remodeling, labor, or birth of viable pups [29,30]. The proposed mechanism for functional progesterone withdrawal in murine and human pregnancy for cervix remodeling and labor is provided in Figure 1. Evidence for progesterone to maintain pregnancy and the loss of efficacy to sustain an unripe cervix are common characteristics among mammalian models and findings in women. For labor and birth in rodents at term, progesterone declines in circulation—an actual withdrawal of support to maintain pregnancy compared to evidence in women where high concentrations remain until after delivery.

At present, the only FDA-approved drug for the prevention of PTB is the synthetic progestin, Makena^®^, which is indicated for women with a singleton pregnancy and a history of a PTB in a previous pregnancy. This treatment is currently being reevaluated due to efficacy concerns in confirmatory trials. Another comparable therapy, the vaginal progesterone gel Crinone^®^, has failed to receive FDA approval because its efficacy in preventing PTB was not significantly greater than a placebo [7]. Thus, there is opportunity for a pharmaceutical approach to provide an alternative to mechanical interventions such as a cervical cerclage or pessary [31,32,33].

### 2.2. Estradiol for the Treatment of Preterm Cervix Remodeling and PTB

Estradiol is the other major steroid hormone produced by the ovary and, in higher mammals, placenta during pregnancy. In contrast to progesterone, increased circulating estradiol precedes the increase in uterine contractility near term [34]. There is little evidence to suggest that estradiol treatments can solely maintain pregnancy. Rather, estradiol promotes the increased presence of immune cells in the uterus and appears to antagonize anti-inflammatory actions of progesterone [35]. As part of a local inflammatory process that involves prostaglandins, estradiol facilitates contractile activity through the shift to a contractile phenotype by the myometrium [36,37]. These findings raise the possibility that estradiol may be part of the mechanism associated with cervix dilation and effacement rather than the previous phase of ripening.

### 2.3. Regulation of Structural and Proinflammatory Processes for Cervix Remodeling

Since parturition, whether at or before term, mirrors a pro-inflammatory process, theoretically, therapy directed at down-regulating pro-inflammatory cytokines and paracrine signaling in the utero-cervico-vaginal region could prevent premature cervix ripening and delay preterm labor and block increased uterine contractions that lead to PTB. Regulating the generation of reactive oxygen species (ROS) or expression of matrix metalloproteinases (MMPs) that degrade collagen cross-linking in the extracellular matrix could also accomplish this objective. MMPs, MMP-2, and MMP-9, in particular, play a role in the prepartum cervix ripening process at term [38] and the overexpression of MMPs has been implicated as a potential biomarker in predicting PTB [39]. Theoretically, MMP inhibitors are also possible therapeutic candidates to forestall prepartum cervix ripening or PTB. Alternatively, local treatments that promote MMPs may facilitate ripening as a complement to induction of labor in women with dystocia or prolonged pregnancy. Indeed, MMP inhibitors have been used to treat diseases in various animal models [40]. However, such treatments fail to block metastatic growth of tumors in clinical trials [41], and effects of MMP inhibitors on physiological remodeling processes like cervix remodeling remain to be determined.

The prospect of focusing a therapeutic pharmaceutical approach on local targets also has merit to mitigate tissue-specific proinflammatory activities. Characteristically, reactive oxygen species (ROS) in or near the cervix activate nuclear factor kappa light chain enhancer of activated B cells (NF-κB) transcription pathways via the stimulation of pro-inflammatory receptors [42,43]. This in turn induces pro-inflammatory cytokines, such as IL-1 and TNF-α, and downstream activation of MMPs [44,45]. Therefore, ROS activity in the cervix presents an upstream target for lessening the overexpression of MMPs seen in PTB. In fact, a clinical pilot study found that vaginal administration of the glycoprotein lactoferrin reduced oxidative stress in amniotic fluid of pregnant women [46].

Macrophages that reside in the cervix and increase in numbers before term are known to produce many proinflammatory cytokines, prostaglandins, and collagen-degrading enzymes that are associated with ripening [26]. However, in rodents, PR is only found in the stroma and epithelial cells of the cervix and adjacent tissues during pregnancy, but not in resident macrophages [26,29], while the cellular location of PR has yet to be determined in women. These findings raise the possibility that cells other than macrophages mediate progesterone’s function in sustaining pregnancy. As paracrine factors regulate local inflammatory processes by resident immune cells, both stromal cells and macrophages may be the focus of treatments that control cervix ripening in preparation for birth.

The consequence of mitigating the production of proinflammatory processes or macrophage activities may reduce the risk of PTB. By example, cervical length is correlated with a high risk of PTB. Typically diagnosed by transvaginal ultrasound, cervix length of less than 25 mm midway through pregnancy is correlated with increased incidence of spontaneous PTB [13,47,48,49] and a criterion for consideration of therapeutic intervention. Moreover, vaginal administration of progesterone in pregnant women with short cervical lengths reduced the rate of PTB by 45% [50]. Although use of progesterone in clinical practice to reduce the incidence of PTB remains unresolved, local treatments that minimize systemic consequences hold promise.

## 3. Advantages of Nanoformulations to Reduce Risks of Premature Cervix Remodeling and Forestall PTB

### 3.1. Surmounting Physiological Barriers to Drug Delivery

The multitude of barriers that obstruct drug delivery to a target tissue following administration of a therapeutic substance has led to the advancement of nanoparticles as carriers in drug delivery systems [51]. The obstacles in drug delivery can be attributed to the biological barriers associated with the organism, such as tissue barriers, cell trafficking, and the immune system [52,53], along with the physiochemical properties of the drug, such as optimal pH range, molecular weight, and lipophilicity [54]. Impaired drug delivery can result in poor efficacy, low bioavailability, and toxic effects [55,56]. The multidisciplinary field of nanotechnology has not only been shown to address drug delivery issues but also to provide sustained and controlled release of a therapeutic substance at the target tissue [57]. Despite the promise of nanomedicine, its implementation and availability to patients has lagged [58,59]. Some of the factors that account for this translational gap include the discrepancy of results between animal and human studies [60] and lack of universal standards [61] and implementation of proper quality control measures in a clinical setting [62]. As the translational gap is addressed, studies continue to show the promise of utilizing nanomedicine to manage a wide range of diseases such as cancer, atherosclerosis, and Alzheimer’s disease [62,63,64]. These diseases each come with their own multitude of biological barriers that need to be overcome, and the use of nanotechnology for the management of this broad spectrum of ailments is a testament to the versatility and wide variety of nanoparticle platforms that can be designed today [52].

### 3.2. Approaches to Circumvent the External Maternal Interface

The cervico-vaginal mucus layer produced by epithelial cells lining the vagina and cervix is a significant physiological barrier affecting drug delivery and treatment efficacy [65]. Rich in glycoproteins, lipids, and an inherently unique microbiome [66,67], the mucus layer is a double-edged sword in that it presents an immunological and physical barrier that prevents both pathogens and therapeutic drugs from coming in contact with vaginal epithelial cells. Muco-adhesion is a formulation strategy that has been utilized to prolong the residence time of a drug when administered to the vagina or any other tissue that contains a mucus membrane [68]. Muco-adhesive formulations vary as they can take advantage of any number of the physio-chemical properties of the vaginal mucus layer. For example, one study showed that nanocapsules synthesized from polymers possessing cationic properties showed more adhesiveness to a mucosal surface when compared to nanocapsules synthesized from polymers possessing anionic properties [69]. This finding could be due to the potential interactions of the negatively charged mucins with the positively charged polymers [70]. However, muco-adhesive formulations are not without their limitations. Muco-adhesive nanoparticles can become trapped in the outermost layer of the protective barrier and their ability to penetrate through the mucus layer into the epithelial cells can be hindered [71]. Furthermore, as mucus is continuously secreted and cleared, the turnover rate of mucus in the cervico-vaginal tract will affect the mean residence time of muco-adhesive nano-formulations. In recent years, there has been a paradigm shift in formulation strategies for vaginal administration where mucus-penetrating nanoparticles are being touted as more efficient drug delivery systems than muco-adhesive nanoparticles [72]. In one study, it was shown that mucus-penetrating nanoparticles coated with high-density, low molecular weight polyethylene glycol could reach the deepest layers of the mucus and uniformly distribute their cargo over the vaginal epithelium, while muco-adhesive nanoparticle formulations, on the other hand, aggregated in the mucus layer, leading to poor distribution of their cargo over the vaginal epithelium [73,74]. As more nanoparticle platforms are optimized for vaginal administration, issues of toxicity, drug distribution, and bioavailability will gradually be resolved.

### 3.3. The Uterine First-Pass Effect

The vagina has been recognized as an exceptional alternative route for the administration of drugs [75]. Originally, this method of administration was used exclusively for locally acting agents such as steroids, antimicrobials, and spermicidal agents [76,77]. Over time, a plethora of research papers made the case for utilizing this alternative route for the delivery of therapeutics to the organs that make up the female reproductive tract [78]. Several advantages of vaginal administration include the presence of a dense network of blood vessels, large surface area, ability to circumvent hepatic first-pass metabolism, and ability to utilize the so-called “uterine first pass effect” [76,79,80]. The advent of the uterine first pass can be traced to a study that compared intramuscular and vaginal administration of progesterone [81] and showed that vaginal administration enhances delivery to the uterus. Since then, several studies have shown preferential distribution of drugs to the uterus when administered vaginally [75,82,83,84]. The uterine first pass effect is a result of multiple physiological and biochemical mechanisms working concurrently [83]. The hallmark of the uterine first pass effect is the targeted delivery of a drug to the uterus or cervix accompanied by low levels of said drug in the systemic circulation [84], allowing for heightened efficacy and minimal adverse effects [75,85]. With a name like uterine first pass effect, a naïve researcher might instinctively draw parallels to the hepatic first pass effect associated with drugs taken orally, which is not entirely wrong. Both phenomena reduce the amount of drug reaching the systemic circulation, but an important distinction is the extent of metabolism (or lack thereof) associated with both routes. While the liver contains a high number of drug-metabolizing enzymes [86], the vagina has relatively low enzymatic activity [80]. Sometimes a drug is administered vaginally rather than orally just for this reason to avoid extensive metabolism of the drug via the hepatic first pass effect [79]. 

### 3.4. Limitations of Vaginal Administration

As with any route of administration, drugs introduced vaginally have several obstacles to overcome in order to produce desired clinical outcomes [79,85]. Even though the vagina is permeable to a wide range of compounds, especially low molecular weight compounds [87], not all drugs conform to these optimal physiochemical properties and some of the innate characteristics of the vagina present a challenge that leads to poor absorption and suboptimal efficacy of a given drug. This can include the self-cleansing action of the vaginal tract [88,89], fluctuations in the thickness of the vaginal epithelium [90], and the microbial community that exists in the cervico-vaginal region [91,92] along with a myriad of other physiological factors. Standardized assessments of treatment efficacy with respect to structural and physiological biomarkers at the mucus-epithelial vaginal interface, vaginal microbiome, and composition of cervical-vaginal fluid are essential to ensure therapeutic objectives.

## 4. Vaginal Nanoformulations to Block Premature Cervix Remodeling and Prevent PTB

### 4.1. Advantages of Vaginal Nanoformulations

The therapeutic effect of a drug can only be exerted if the delivered dose reaches the target site in relevant concentrations. Oral and parenteral delivery result in high levels of systemic exposure compared to targeted delivery. Further, low solubility, permeability, and first pass effect (oral route) limit bioavailability. Regardless of how potent a molecule might be for a given indication, selection of an appropriate delivery system is crucial to avoid metabolic degradation and provide therapeutic concentrations of the drug at the target site for an effective period of time [93]. As molecules are incorporated into nanoparticles, they adopt pharmacokinetic properties of the carrier. Pharmacological properties such as absorption into blood, distribution, metabolism, and liberation in a timely manner may be affected. Through proper formulation, these properties may be modified to aid in extending the half-life of a drug, reducing the drug dose, and minimizing off-target effects [94,95,96]. Numerous reports discuss the ease of maternal-to-fetal transfer of drugs using nanoparticle technology, including different nanoformulation categories such as lipid-based drug delivery systems, polymeric nanoformulations, dendrimers, etc. [97,98,99]. Localized delivery of these nanotherapeutics could enhance the efficacy of drugs for PTB by delivering high concentrations at the uterine site. 

The key challenges in delivering molecules to prevent PTB by the vaginal route are: (1) solubility in the vaginal milieu and (2) permeation across the cervico-vaginal mucosa. Both these issues can be overcome by formulating the molecules in surface-modified nanoparticles. Based on the physiological environment of the vagina and the obstacles that can interfere with nanoparticles, various modifications have also been explored [96,100,101].

As the majority of drugs are highly lipophilic and possess poor aqueous solubility, the most favorable approaches for drug administration include oral and intravenous routes [93]. The commonly employed class of molecules for PTB include antenatal corticosteroids and tocolytics (including calcium channel blockers, beta-adrenergic receptor agonists, and nonsteroidal anti-inflammatory drugs, or NSAIDs) that belong to BCS class II or IV, which makes their conveyance to the delivery site even more problematic given the existence of numerous biological barriers [95,102]. It is challenging to solubilize an efficacious dose of a drug in 1–2 mL of vaginal fluid versus 250 mL of stomach fluid. Thus, use of an advanced formulation technology is essential to solubilize drug molecules in such a limited fluid volume. 

Mucus also presents a permeability barrier for drug molecules. Mucin proteins in the cervico-vaginal mucus layer are decorated with hydrophobic and electrostatic regions that lead to adhesive interactions with virions and charged particles. Using charged nanoparticles or polymers could facilitate bioadhesion or mucopenetration to assist in long-term delivery of a drug to its target site [103]. Moreover, to avoid daily administration of therapeutics and for better patient compliance, nanoparticles loaded in vaginal rings or inserts could be a potential strategy for patients at risk for PTB [104]. Other approaches including the use of pH sensitive nanocarriers targeting relevant receptors (e.g., oxytocin) and microneedle technology have been widely explored for different routes of administration, including vaginal, and warrant investigation for preventing PTB via the vaginal route [105,106].

### 4.2. Novel Approach to Anti-Inflammatory Therapeutics

Taking into consideration the advantages of vaginal administration and utilizing nanoparticle platforms to overcome barriers associated with this route of administration, there are several studies that have tested vaginal nanoformulations of therapeutics in preclinical trials in order to treat PTB. In a previous study, the Reznik lab had shown that inhibiting sphingosine kinase can reduce the incidence of PTB in an inflammation-induced mouse model [107]. Building on their lab’s previous work, they developed a vaginal nanoformulation of their sphingosine kinase inhibitor (SKI II) and tested its efficacy in inflammation-induced Swiss Webster mice [108]. To circumvent the solubility issues associated with SKI II, they loaded their drug into a vaginal self-nanoemulsifying drug delivery system (SNEDDS). Composed of oil, co-solvent, drug, and surfactant [109], SNEDDS have been a novel nanoparticle platform in the pharmacotherapeutic field due to their ability to increase the bioavailability and solubility of lipophilic compounds [110,111]. In these studies, the physiochemical properties of the nanoparticle platform were optimized to ensure sufficient delivery of SKI II to the uterus and minimize systemic circulation of their drug thus minimizing adverse effects. Their research showed that SKI II-loaded SNEDDS significantly increased the number of pups rescued from lipopolysaccharide (LPS) induced PTB when compared to mice treated with blank-loaded SNEDDS and mice treated with just LPS + Phosphate Buffered Saline [108]. Interestingly, the blank-loaded SNEDDS by itself possessed the ability to rescue pups from being spontaneously aborted when compared to the group treated with LPS + Phosphate Buffered Saline (although not statistically significantly). This could be due to one of the components in their SNEDD nanoparticle, N,N-dimethylacetamide (DMA). The Reznik lab has shown that the commonly used solvent DMA ameliorates the proinflammatory response associated with LPS-induced PTB in murine models [112]; current investigations for a vaginal nanoformulation of DMA are ongoing in their lab. The versatility of SNEDDS nanoparticle platforms is further indicated in another study performed by the Reznik lab to demonstrate the efficacy of a vaginal tablet formulation of 17-α Hydroxyprogesterone caproate (17P) in inflammation-induced PTB [113]. As in the previous SKI II study, the physiochemical properties of their nanoparticle platform were optimized to promote absorption of their 17P-loaded vaginal tablet. The 17P-loaded SNEDDS were able to delay the onset of labor and reduce the incidence of PTB in LPS-stimulated mice [Figure 2]. Comparing the survivability curves from both the SKI II-loaded SNEDDs study and the 17P-loaded SNEDDs study, a higher percentage of pups were rescued from PTB with the SK II-loaded SNEDDs possibly suggesting that it is a more efficacious therapeutic agent than the 17P-loaded SNEDDs.

### 4.3. Dual Therapy Study

The value of a local approach to potentially treat premature cervix ripening and early onset of labor is also illustrated by a recent study by Zierden et al. [114]. Contrary to the previous Reznik study with results of vaginal 17P, the progesterone gel Makena^®^ or Crinone^®^ (P4) did not alter inflammation-induced PTB in mice. However, the combined treatment of P4 with a histone deacetylase inhibitor (HDAC) in a nanosuspension reduced the incidence of PTB by 50%, likely a consequence of the recognized anti-inflammatory properties of HDAC inhibitors [115,116]. Thus, the use of the HDAC inhibitor Trichostatin A (TSA) could thus serve to amplify anti-inflammatory effects. A critical feature of studies from both the Reznik and Ensign labs is optimization of the physiochemical characteristics of the nanoparticle platform for vaginal administration. Future translational work thus needs to include optimization of treatment along with assessment of appropriate biomarkers for the study of cervix ripening as essential components to evaluate the efficacy to prevent unintended maternal–fetal consequences and PTB. By example, the role played by prostaglandins in cervix remodeling following induction of labor is complicated by unintended systemic side effects, indirect compound endpoint (Bishop score and vaginal birth), and multiple vignettes in clinical practice [117]. These considerations are part of the challenge to understand the mechanism of cervix ripening, the definition and timeline of which remain to be clarified. The growing demand for medical interventions in obstetrical practice over the past decades provides the impetus for research into novel interventions that target local inflammatory processes in the prepartum cervix as an early mechanism preceding labor and parturition. These promising preclinical trials have the potential to bridge the translational gap and the scarcity of effective treatments to prevent preterm cervix remodeling and premature birth.

### 4.4. Evidence for the Efficacy of Vaginal Nanoformulations to Prevent or Forestall PTB

As shown in Table 1, different nanoformulations have been explored to enhance the efficacy of commonly used drugs to prevent PTB. Progesterone, a corticosteroid, has been effectively delivered using unique approaches through the vaginal route. A report by Cam et al. discusses the development of progesterone-loaded poly(lactic) acid (PLA) fibrous polymeric patches using electrospinning and a pressurized gyration technique [118]. Patches prepared using pressurized gyration had higher production yields and tensile strengths than electrospinning; therefore, this technique was the preferred method. The optimized patches did not affect viability, cell adherence, or proliferation of Vero 6 cells and therefore were highly biocompatible for vaginal use. The progesterone-loaded PLA patches had melting points above body temperature, thereby evading the risk of melting. In vitro release yielded ~20% release within 210 min in comparison to ~12% from the commercial progesterone-loaded pessaries. Complete drug release from the patches was observed within 24 h. Results from in vitro bath experiments showcased a decrease in myometrial contractions in both KCl and cumulative (−)-noradrenaline treated pregnant rat uteri with both vaginal patch and oral progesterone treatment. No significant difference was observed between the patch and oral progesterone groups, thereby showing that high bioavailability, reduced dosage frequency, and minimal side effects were achievable with in vivo administration of the patch [118]. A similar report by Brako et al. investigates the applicability of bioadhesive nanofibers loaded with progesterone for PTB [119]. Polyethylene oxide (PEO) and carboxymethyl cellulose (CMC) were used as the carrier matrix due to their mucoadhesive and sustained release properties. Uniform blends of PEO, CMC, and progesterone were obtained and later sonicated using a probe sonicator. The nanofibers were prepared using a pressurized gyration technique at ambient temperatures with 400 nm diameters and up to 25% progesterone loading. Thermal and spectroscopic measurements confirmed that some of the drug was entrapped in a crystalline state. Release kinetics from the nanofibers were similar to those of the commercial pessary [120]. Similarly, a detailed characterization of a thermoreversible gel prepared using Pluronic F127, a nonionic amphiphilic triblock copolymer based on concentration, has been studied to create an in situ formulation that gels upon vaginal delivery for prolonged drug release [121]. Nanosuspensions of progesterone were prepared using wet-milling techniques in the presence of different surfactants and studied for their viscoelastic properties and in vivo gelation using a fluorescent dye. In addition, a delivery platform of salbutamol sulfate using bioadhesive polymers showed a strong positive correlation (0.91) between in vitro and ex vivo permeation, suggesting that salbutamol vaginal bioadhesive tablets could be optimized for in vivo performance [122]. Nanoformulation technology could help to enhance the efficacy of such drug-loaded bioadhesive tablets. As previously stated, due to the uterine first-pass effect, vaginally administered formulations can reach the uterus before being transported to the systemic circulation. Finally, a study conducted by Correia et al. examines the role of a nanostructured lipid carrier in delivering progesterone via a pessary formulation [120]. The formulation was found to be highly biocompatible with human immortalized keratinocytes (HaCaT) cells at a concentration of 10 µg/mL. Sustained drug release was observed for up to 24 h from the formulation as opposed to the neat drug [120].

## 5. Summary and Conclusions

PTB, defined as birth before 37 completed weeks of gestation, causes more perinatal morbidity and mortality than any other obstetrical complication or clinical disorder [1,2,3]. Data collected by the WHO confirm that the annual rate of PTB worldwide has not improved for many years and is greater than 10% in most countries [4]. While PTB rates are generally higher in the developing world, among industrialized nations, the US has the highest rate of PTB [2]. In the US particularly, because advances in neonatology have occurred more rapidly than in the field of obstetrics, more and more extremely premature infants survive the neonatal period but sustain permanent neurologic injury. The acute sequelae of PTB, such as respiratory distress syndrome and necrotizing enterocolitis, are followed by lifelong respiratory, metabolic, and neurologic abnormalities, including retinopathy of prematurity and cerebral palsy [6]. The costs of PTB for individuals, families and society are staggering.

Currently, the US Food and Drug Administration is calling for withdrawal of the only approved drug for the prevention of PTB, namely hydroxyprogesterone caproate, also known as Makena, which was shown to have no significant efficacy compared to a placebo in the PROLONG trial [123]. The most successful drug therapy to prevent PTB is vaginal progesterone, which is administered off-label to women who have had a PTB before and are therefore at risk for delivering preterm. Vaginal progesterone, however, has only a modest effect on the incidence of PTB [16]. As far as tocolytics, magnesium sulfate, indomethacin, and nifedipine, which are the three most often used, only postpone PTB for approximately 48 h [95], which allows time for the administration of antenatal corticosteroids for the promotion of lung maturity but does prolong gestation long enough to significantly effect neonatal or long-term outcomes. As for beta mimetics, ritodrine and terbutaline were originally thought to hold much promise as agents to prevent PTB but are no longer used due to concerns about their toxicity.

Historically, drug development for PTB has been hindered, not only by concerns about maternal toxicity, but by fears of teratogenic effects on the fetus. Both types of adverse effects result from drugs meant to act in either the cervix or the uterus having off-target actions. A convenient first approach to avoiding these complications is the vaginal route of administration. Advantages of administering drugs vaginally, as outlined in this review, include the presence of a highly vascularized mucosa, a large surface area, and the avoidance of hepatic first-pass metabolism [85,86,87,88]. Above all, vaginal administration benefits from the uterine first-pass effect, which allows drugs introduced into the vagina to be transported by the vaginal–cervical–uterine portal vascular system, permitting high concentrations of vaginally administered drugs to reach target sites in the cervix and uterus, while low concentrations reach the systemic circulation [76,77,78,79,80]. Still, as described in this review, there are several potential barriers related to the vaginal route of administration. These may include poor absorption [88,89], unfavorable effects on the vaginal microbiome [91,92], and the cervico-vaginal mucous layer—a thick natural barrier rich in glycoproteins and lipids preventing drugs introduced in the vaginal cavity from reaching their target sites in the gynecologic tract [79]. 

The most exciting development to overcome these challenges is the revolution in pharmacology and pharmaceutics known as nanomedicine [124]. Drugs loaded into nanoparticles acquire the pharmacokinetic properties of their carriers. Absorption, distribution, metabolism, and excretion may all be altered, as the drug cargo disguises its own unfavorable characteristics for the more beneficial ones of the nanoparticle. Drugs with promising efficacy in vitro, but poor results in vivo, because of their failure to accumulate at therapeutic concentrations at target sites, can now be re-visited as potential effective pharmacotherapy. Vaginally administered nanoparticles loaded with drug cargo, specifically engineered for mucoadhesion or mucopenetration, can overcome the cervico-vaginal mucous barrier and be directed to the target tissue with reduced systemic drug levels and reduced penetration of the placental barrier [103]. The utility of vaginal nanoformulations to prevent or forestall PTB has now been explored in several trials as outlined in this review. 

We have shown that a vaginally administered self-nanoemulsifying drug delivery system (SNEDDS) can delay the onset of inflammation induced PTB in vivo in two separate studies [108,113]. After we found that SKI II, a sphingosine kinase inhibitor, could prevent PTB when administered intraperitoneally [107], we tested a SKII-loaded SNEDDS composed of oil, co-solvent, and SKI II drug cargo, and found that it rescued a significant number of pups from PTB in lipopolysaccharide-induced mice [108]. In a second study, we reported similar findings, using a nanovaginal formulation loaded with 17-alpha hydroxyprogesterone caproate (also known as Makena) [113]. Work aimed at testing a vaginal nanoformulation of N,N-dimethylacetamide, a widely used drug excipient which we have shown to be a candidate for re-purposing for PTB [112,125,126], is ongoing. Ensign et al. have recently reported that a vaginal nanoformulation carrying the histone deacetylase inhibitor Trichostatin A combined with progesterone as its drug cargo decreased inflammation-induced PTB in their murine model by 50% [114].

The ability to deliver therapeutic concentrations of drug cargo to the cervix and uterus, while maintaining very low concentrations in the maternal systemic circulation and preventing significant concentrations from reaching the fetus, heralds a new era in drug development for PTB. Nanomedicine raises the possibility of re-visiting drugs previously considered too toxic, as well as testing new drugs that would otherwise not be candidates for obstetrical disorders. Efforts in translational research for PTB should be re-directed and focused on vaginal nanoformulations.

## 6. Patents

S.E.R. has a patent pending (13/536/946), titled “Administration of N,N-dimethylacetamide and its monomethylated metabolites for the treatment of inflammatory disorders”.

## Figures and Tables

**Figure 1 pharmaceutics-14-02019-f001:**
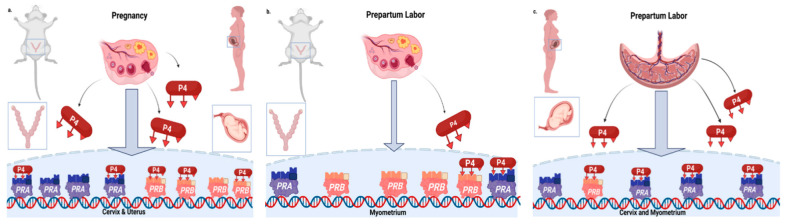
**A cross-species comparison of the importance of progesterone (P4) for pregnancy and parturition.** (**a**) In mice and women, sustained increasing production of progesterone (P4) from the corpus luteum of the ovary is essential to establish and maintain pregnancy. P4 production in women shifts to the placenta in the first trimester of pregnancy. Most physiological effects of P4 are mediated by two classic genomic nuclear progesterone receptor isoforms, PR-A and PR-B, respectively. (**b**) P4 declines in the circulation just before labor in prepartum mice. This period comes after the cervix ripens while P4 concentrations are near or at peak [24]. (**c**) In prepartum women, P4 remains near peak or increases in the circulation until birth occurs. Thus, cervix ripening and labor reflect a functional withdrawal of the effects of P4 for parturition in women, while in mice there appears to be a functional loss of response to P4 for ripening of the cervix followed by an actual loss of P4 for labor and birth. Adapted from “*Molecular Pathway and Mouse Phenotype* (*Layout*)”, by BioRender.com (accessed on 5 September 2022). Retrieved from https://app.biorender.com/biorender-templates (accessed on 20 June 2022).

**Figure 2 pharmaceutics-14-02019-f002:**
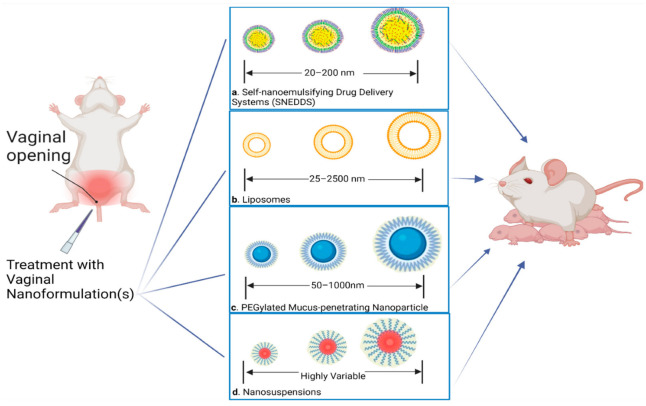
Schematic overview of different nanoparticle platforms that have been studied for their ability to improve drug delivery to the female reproductive tract and reduce the incidence of PTB. Nanoformulations significantly improve the solubility of active pharmaceutical ingredients in the vaginal milieu and facilitate uterine delivery. Lipid-based nanoformulations include (**a**) spontaneously forming nanoemulsions and (**b**) liposomes. Self-nanoemulsifying systems are composed of vegetable oil + non-ionic surfactant while liposomes are composed of phospholipids. (**c**) Polyethylene glycol coated nanoparticles rapidly cross the cervico-vaginal mucus layer compared to non-coated nanoparticles and drug molecules alone. (**d**) Nanosuspensions contain nano-sized drug particles (<1000 nm) and biocompatible stabilizers. Top-down, i.e., high-energy milling or bottom-up, i.e., solvent evaporation techniques are commonly used to prepare nanosuspensions of drug molecules. Created with BioRender.com (accessed on 5 September 2022).

**Table 1 pharmaceutics-14-02019-t001:** Nanoformulations of Sphingosine Kinase II (SKII) inhibitor, 17-α hydroxyprogesterone (17OHP), and Progesterone (P4) for the prevention of PTB.

Active Compound	Delivery System	Preparation Method	Particle Size (nm)	Key Findings	Ref.
** *SKII inhibitor* **	Self-nanoemulsifying system (SNEDDs)	-	36.78	>500-fold increase solubility Reduced PTB No teratogenicity	[108]
** *17OHP* **	Solid self-nanoemulsifying preconcentrate (S-SNEDDS) loaded vaginal tablet	Adsorption with direct compression	49.55 ± 2.7	Higher dissolution rate vs. API alone >50% inhibition of TNF-α release Higher PTB rate, longer time to deliver	[113]
** *P4* **	Mucoinert nanosuspension	Wet milling	~250	Improved delivery Higher biocompatibility Enhanced P4 AUC in cervix, uterus	[103]
** *P4* **	Poly(lactic) acid fibrous polymeric patches	Pressurized gyration	7600–7900	~97% drug encapsulation Faster drug release patch vs. pessary No toxicity in vitro	[118]
** *P4* **	Bioadhesive nanofibers	Pressurized gyration	400	25% *w*/*w* P4-loaded nanofibers same drug release from nanofibers and commercial product	[119]
** *P4* **	Nanostructured lipid carriers (NLC) in pessaries	Melt emulsification with ultrason-ication	316 ± 0.01	>96% drug encapsulation efficiency Biocompatible 24 h sustained release	[120]
** *P4* **	Nanosuspension loaded in hydrogel	Wet milling nanocrystal synthesis	233–380	Pluronic F127 hydrogel improved elasticity vs. Crinone^®^ reduced PTB rate	[121]

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
