# Peer review of "Vaginal Nanoformulations for the Management of Preterm Birth"

_pharmaceutics, 2022, doi:10.3390/pharmaceutics14102019_

Round 1

Reviewer 1 Report

The manuscript "Vaginal Nanoformulations for the Management of Preterm Birth" is very interesting and well-organized. I strongly support its publication. I would recommend to revise the English style and format.

Author Response

The manuscript "Vaginal Nanoformulations for the Management of Preterm Birth" is very interesting and well-organized. I strongly support its publication. I would recommend to revise the English style and format.

We thank Reviewer 1 for these kind words. We have reviewed the entire article and improved the writing style wherever possible.

Reviewer 2 Report

This review is dealing with the novel therapeutic approaches for preterm birth. It is an interesting manuscript. However, the literature search strategies are not described. I have the following comments:

1.            What search strategies were used for this review?

2.            Which databases were searched?

3.            Is this a systematic review?

4.            What are the reasons for the high rate of preterm births in the US? The authors should further discuss this topic in the introduction section.

5.            What are the reasons Crinone gel did not obtain FDA approval?

6.            What are the indications for the use of Makena?

6.           What are the recommendations for the management of short cervix (Ë‚25 mm between 16+0 and 24+0) in the US?

Author Response

This review is dealing with the novel therapeutic approaches for preterm birth. It is an interesting manuscript. However, the literature search strategies are not described. I have the following comments:

  1. What search strategies were used for this review?

The PubMed database was searched using the following keywords: "vaginal administration", "vaginal progesterone", "preterm birth", "cervical ripening/cervical remodeling (several of Dr. Yellon's papers came up), "inflammation + preterm birth", "makena", "nanoformulations + mucoadhesive" and "nanoformulation + mucus-penetrating" and accessing literature from 1979 to the present. A comment about our search strategy has been added at the top of Page 2.

  1. Which databases were searched?

Please see the response to Point 1.

  1. Is this a systematic review?

Yes, this article is a systematic review. We have added a comment stating the same at the top of Page 2.

  1. What are the reasons for the high rate of preterm births in the US? The authors should further discuss this topic in the introduction section.

We thank Reviewer 1 for this excellent question. This is a complex issue. The high rate of preterm birth in the US is believed to be related to the high rate of obesity and cardiovascular disease among US women as well as social determinants of health in this diverse population. A statement has been added to the introduction section reflecting these factors on Page 1. Also, a citation has been added to support the statement.

  1. What are the reasons Crinone gel did not obtain FDA approval? 

Crinone gel did not obtain FDA approval because it had no greater efficacy in preterm birth prevention than a placebo. A statement reflecting this point has been added to the manuscript on Page 3.

  1. What are the indications for the use of Makena?

Makena is indicated for women with a singleton pregnancy who have a history of preterm birth in a previous pregnancy. This information has been added to the manuscript on Page 3.

  1. What are the recommendations for the management of short cervix (Ë‚25 mm between 16+0 and 24+0) in the US?

Currently, there are two recommended treatment options for the management of short cervix (<25 mm) occurring between 16 and 24 weeks’ gestation in the US: cerclage and vaginal progesterone.  A statement reflecting this information has been added to Page 2.

Round 2

Reviewer 2 Report

All comments have been addressed and I 

believe the manuscript is now suitable 

for publication.